Oral bacteriophages: metagenomic clues to interpret microbiomes

Banar Maryam 1
Rokaya Dinesh 2
Azizian Reza 3 4
Khurshid Zohaib 5 6
Banakar Morteza dr.mbanakar@gmail.com 7
1 Department of Pathobiology, School of Public Health, Tehran University of Medical Sciences , Tehran , Iran
2 Department of Basic Medical and Dental Sciences, Faculty of Dentistry, Zarqa University , Zarqa , Jordan
3 Biomedical Innovation and Start-up student association (Biomino), Tehran University of Medical Sciences , Tehran , Iran
4 Pediatric Infectious Diseases Research Center (PIDRC), Tehran University of Medical Sciences , Tehran , Iran
5 Department of Prosthodontics and Implantology, College of Dentistry, King Faisal University , Al-Hofuf, Al Ahsa , Saudi Arabia
6 Center of Excellence for Regenerative Dentistry, Department of Anatomy, Faculty of Dentistry, Chulalongkorn University , Bangkok , Thailand
7 Dental Research Center, Dentistry Research Institute, Tehran University of Medical Sciences , Tehran , Iran
Souza Valeria
Electronic publication date: 2024 Feb 20
Publication date: 2024
Volume: 12
Electronic Location ID: e16947
Received 2023 Nov 14; Accepted 2024 Jan 24
Copyright: ©2024 Banar et al.
Copyright year: 2024
Copyright holder: Banar et al.
License: This is an open access article distributed under the terms of the Creative Commons Attribution License, which permits unrestricted use, distribution, reproduction and adaptation in any medium and for any purpose provided that it is properly attributed. For attribution, the original author(s), title, publication source (PeerJ) and either DOI or URL of the article must be cited.
License URL: https://creativecommons.org/licenses/by/4.0/

Keywords: Oral microbiome, Oral bacteriophage, Oral phagoeome, Metagenomics

Funding: Zarqa University, Jordan This research is supported by the Zarqa University, Jordan. The funders had no role in study design, data collection and analysis, decision to publish, or preparation of the manuscript.

==============================
Bacteriophages are bacterial viruses that are distributed throughout the environment. Lytic phages and prophages in saliva, oral mucosa, and dental plaque interact with the oral microbiota and can change biofilm formation. The interactions between phages and bacteria can be considered a portion of oral metagenomics. The metagenomic profile of the oral microbiome indicates various bacteria. Indeed, there are various phages against these bacteria in the oral cavity. However, some other phages, like phages against Absconditabacteria, Chlamydiae, or Chloroflexi, have not been identified in the oral cavity. This review gives an overview of oral bacteriophage and used for metagenomics. Metagenomics of these phages deals with multi-drug-resistant bacterial plaques (biofilms) in oral cavities and oral infection. Hence, dentists and pharmacologists should know this metagenomic profile to cope with predental and dental infectious diseases.

Introduction

In 1958, an American molecular biologist Joshua Lederberg coined the word “microbiome” to define the ecological population of commensal, symbiotic, and pathogenic microbes that coexist in our bodies (Lederberg & McCray, 2001). In other words, the term “microbiome” includes all microbes residing in the body, their genomes, and ecosystems (Zarco, Vess & Ginsburg, 2012). These microorganisms have different habitats, including the oral cavity, skin, gastrointestinal, urogenital, and respiratory tracts (Zarco, Vess & Ginsburg, 2012; Di Stefano et al., 2022; Sharma et al., 2018). Each site has a distinct microbiome that differs in function and components (Bacali et al., 2022).

The oral microbiota consists of different microbial species, including bacteria, protozoa, archaea, viruses, fungi, and ultra-small organisms (candidate phyla radiation group) (Cornejo Ulloa, Van der Veen & Krom, 2019; Zaura et al., 2014; Wade, 2021; Radaic & Kapila, 2021). The oral microbiome has an undeniable role in a person’s nutritional, physiological, and immune system development (Caselli et al., 2020). Various functions of the microorganism are involved in maintaining healthy oral health. For example, commensal microorganisms prevent the colonization of pathogenic microbes in the oral cavity by competing for colonization sites. In addition, the oral microbiota produces bacteriocins that kill pathogens and have anticancer properties. Moreover, the oral microbiome has a role in systemic nutrient cycling concerning nitrate metabolism (Sedghi et al., 2021).

The disturbances in the composition and number of oral microbiomes result in oral diseases (Bacali et al., 2022). So far, the relationship between oral microorganisms and some oral diseases has been identified, including periodontitis, dental caries, alveolar osteitis, peri-implantitis, tonsillitis, endodontic infections, oral cancers, and mucosa diseases such as leukoplakia and lichen planus (Duran-Pinedo & Frias-Lopez, 2015; Gao et al., 2018). Because the mouth is the main entryway of the body, it may transfer pathogenic and commensal microorganisms to the adjacent body parts, thereby contributing to the development of systemic illnesses (Girija & Ganesh, 2022). Studies showed a link between oral infections and some systemic ailments, including pneumonia, Alzheimer’s disease, preterm birth, polycystic ovary syndrome (PCO), ictus, stroke, obesity, inflammatory bowel disease, cardiovascular disease, liver cirrhosis, and diabetes (Duran-Pinedo & Frias-Lopez, 2015; Gao et al., 2018). Interestingly, dysbiosis of the mouth microbiota involves the expansion of several specific cancers, including the role of Fusobacterium nucleatum in colon cancer (Yamashita & Takeshita, 2017; Peng et al., 2022), as well as some autoimmune diseases such as Sjogren’s syndrome that are related to the disturbed ratios of Firmicutes/Proteobacteria (Lee et al., 2021).

Bacteriophages (prokaryotic viruses) are predominant in the oral virobiome. They can infect oral bacteria and undoubtedly have the potency to shape the oral microbiome through modifications in the structure and attributes of the bacterial population. However, their role in oral health and disease has yet to be fully understood, and investigations in this field are ongoing (Baker et al., 2017). Recently, the advancement of omics methods such as metagenomics has promoted bacteriophage studies and provided valuable information about the diversification and roles of the phages in the mouth (Edlund et al., 2015).

The oral microbiome is among the most complicated microbial communities in our body and, along with the nasal cavity, gut, vagina, and skin, is of great interest to researchers in the field of the Human Microbiome Project. Until recently, most microbiome studies primarily focused on bacterial populations, and few studies assessed other oral biomes, such as the archaeome, protozoome, mycobiome, and virobiome. Therefore, researchers should pay special attention to these members of the oral microbiome. This article reviewed metagenomics studies that have focused on bacteriophages as a component of the human oral microbiota and examined the role of bacteriophages in oral health and disease.

Survey Methodology

We reviewed articles on metagenomics studies that have focused on bacteriophages as a component of the human oral microbiota and examined the role of bacteriophages in oral health and disease using Google Scholar, MEDLINE/PubMed, Web of Science, and ScienceDirect resources from 2001 to 2023. A total of 70 articles were included in this review. We only included research and review articles in English in this review.

Oral Microbiome

The oral microbiome, or microbiota, refers to the microbial population and genomes residing in the mouth (Zaura et al., 2014; Gao et al., 2018). It was first discovered by a Dutch researcher Antony van Leeuwenhoek in 1674 while observing his dental plaque under his invented microscope (Yamashita & Takeshita, 2017; Patil et al., 2013). Oral microbiome is among the most complicated microbial communities in our body and, along with the nasal cavity, gut, vagina, and skin, is of great interest to researchers in the field of the Human Microbiome Project (Peng et al., 2022).

The oral microbiome ranks second after the gastrointestinal tract in terms of number, complexity, and species diversity because of the variety of ecological niches inside the oral cavity and their optimal conditions (Sharma et al., 2018). The oral microenvironment is divergent and dynamic (Martínez, Kuraji & Kapila, 2021). It consists of different surfaces, including teeth, tongue, non-keratinized cheek mucosa, soft and hard palate, gingival sulcus, mouth floor, tonsils, pharynx, lips, and saliva as shown in Fig. 1 (Lee et al., 2021). These surfaces afford appropriate substrates for various microorganisms’ colonization (Caselli et al., 2020; Baker et al., 2017). In addition, the mouth, and specifically the saliva, offers ideal growth conditions for most microbes, including plentiful nutrients (epithelial debris, food consumed by humans, and byproducts of oral microbiota), constant temperature (37 °C), sufficient humidity, and a stable pH (6.75–7.25) (Sharma et al., 2018; Bacali et al., 2022; Deo & Deshmukh, 2019). Like others, the oral microbiome is site-specific, and different oral habitats have specific microbiota (Aas et al., 2005).

Figure 1 Mouth microbiome and its different niches.

Adapted from Lee et al. (2021).

The first microbes (pioneer species) are transferred from mother to newborn during birth (vertical transmission), which is affected by the type of birth (Bacali et al., 2022; Dominguez-Bello et al., 2010). In natural childbirth, the microorganisms of the mother’s vagina, and in the case of birth through a cesarean section, the mother’s skin flora, are the first colonizers. Studies showed that the oral microbiome of babies born by natural delivery has a higher diversity (Holgerson et al., 2011). The type of infant nutrition (breast milk or powdered milk) further determines the oral microbiome. For example, the oral microbiota in breast-fed babies contains Lactobacillus species, but these bacteria are absent in formula-fed babies (Romani Vestman et al., 2013). Teeth eruption is a turning point in the development of the mouth microbiota because it increases the microbial adhesion surfaces and colonization (Baker et al., 2017). The foremost microorganisms that strike and inhabit the oral cavity are aerobes and obligate anaerobes, such as the members of Lactobacillus, Streptococcus, Neisseria, Veillonella, and Actinomyces genera (Sharma et al., 2018; Deo & Deshmukh, 2019). With the eruption of teeth, the span between the teeth and the gums supplies an appropriate environment for establishing and colonizing anaerobic organisms such as Prevotella spp. and Fusarium spp (Sharma et al., 2018). Up to the age of 3 years, the mouth comprises a complicated population of microorganisms. The loss of deciduous teeth and their substitute with permanent teeth plays an extensive role in modifying mouth habitats and changing the residents of oral microorganisms (Baker et al., 2017). The microorganism of older people is similar to the microbiota of childhood before tooth eruption (Deo & Deshmukh, 2019). Some factors involved in the formation of human oral microbiota from the prenatal period to childhood and adolescence are shown in Fig. 2 (Kaan, Kahharova & Zaura, 2021).

Figure 2 Contributing factors in shaping the human oral microbiome during different life stages from fetus to adulthood.

Adapted from Kaan, Kahharova & Zaura (2021).

Determining the accurate content and diversity of the oral microbiota is very hard since the mouth has persistent exposure to the external environment. Hence, its physical and chemical conditions are variable. For example, the exogenous microorganisms in the air, water, and food influence the microbial community of the mouth. Diet, age, smoking, and kissing can change the population of a person’s oral microorganisms. Other factors affecting the oral microbiome are oral care habits like brushing the teeth and using mouthwash, dental materials used to restore teeth, implants, oral prosthetic devices, systemic diseases, and drugs. Oral infections, host genetic background, gender, changes in sex hormones, and immune responses are also impressive in shaping the oral microbial population (Sharma et al., 2018; Bacali et al., 2022; Xu & Gunsolley, 2014).

Due to the specific conditions in the mouth, including the salivary flow and tongue movement during speaking and food chewing, as well as the variations in shear forces, oxygen level, and energy/nutrient sources in different oral habitats, most oral microbiotas produce biofilms (highly structured surface-associated microbial communities) to attach to the oral surfaces (teeth and gums) and prevent their elimination, as well as accommodate to changes in the oral environment (Roberts & Kreth, 2014; Bowen et al., 2018). Oral biofilm formation establishes where the pellicle covers the tooth surface. The constituents of the pellicle are host-specific molecules such as proteins, agglutinins, and mucins, which bacteria use as receptors. The early colonizers (such as Streptococcus spp., Eikenella spp., Actinomyces spp., and Capnocytophaga spp.) and the middle and late colonizers (such as Prevotella spp., Fusobacterium spp., Actinobacillus spp., Porphorymonas spp., and Eubacterium spp.) utilize these receptors to bind to the tooth surface covered by the pellicle. After sufficient growth and an increase in the bacterial population, bacteria will produce extracellular polymeric substances (EPS), and then a mature biofilm structure will form. The formation of biofilm at the gingival margin is valuable for oral health because its persistent attendance precludes the colonization of pathogens in the mouth. According to studies, altering the prevailing microbial residents of a healthy oral biofilm renders the progression of oral infections. Investigations uncovered the possible role of the microorganisms enclosed in abnormal dental biofilms in inducing several oral diseases, such as periodontitis and tooth decay (Fig. 3) (Lee et al., 2021). Treating biofilm-related infections is another challenge concerning the increased resistance of bacterial biofilms against antibiotics (Roberts & Mullany, 2010).

Figure 3 Bacterial residence of dental plaques (biofilm) and their association with periodontal diseases.

Adapted from Lee et al. (2021).

Composition of the oral microbiome

Bacteriome

Bacteria make up most of the oral microbiome (Nagarajan, Prabhu & Kamalakkannan, 2018). So far, researchers have performed many investigations on oral bacteria and identified about 700 bacterial species in this ecosystem (Duran-Pinedo & Frias-Lopez, 2015; Deo & Deshmukh, 2019). These species are categorized into 185 genera and 12 phyla. The 12 recognized bacterial phyla include Firmicutes, Actinobacteria, Proteobacteria, Fusobacteria, Bacteroidetes, Chlamydiae, Spirochaetes, Chloroflexi, SR1, Synergistetes, Saccharibacteria (TM7), and Gracilibacteria (GN02) (Deo & Deshmukh, 2019).

The three phyla SR1 (Candidatus Absconditabacteria), TM7 (Candidatus Saccharibacteria), and GN02 (Candidatus Gracilibacteria) are members of the Candidate Phyla Radiation (CPR) group. They cover half of the earth’s bacteria and can be found everywhere (Wade, 2021). They have specific features, including ultra-small cell size, archaeal-specific RuBisCO genes, 16S rRNA gene self-splicing introns, and downsized genomes lacking CRISPR/Cas system-related genes. They also lack many metabolic and biosynthetic pathways, such as the tricarboxylic acid cycle, the electron transport chain, amino acid and membrane biosynthesis pathways, and ribosomal subunits (Baker et al., 2017).

At the genus level, the core microbiome is shared by healthy people. Some of the oral bacterial genera are Abiotrophia, Stomatococcus, Peptostreptococcus, Streptococcus, Actinomyces, Eubacterium, Bifidobacterium, Lactobacillus, Corynebacterium, Rothia, Propionibacterium, Pseudoramibacter, Neisseria, Moraxella, Veillonella, Campylobacter, Prevotella, Fusobacterium, Capnocytophaga, Desulfobacter, Treponema, Eikenella, Desulfovibrio, Leptotrichia, Hemophilus, Wolinella, Selemonas, and Simonsiella (Deo & Deshmukh, 2019). These bacterial genera are highly diverse, encompassing both gram-positive and gram-negative cocci and rods. Some of them are rare, such as organisms of the genus Abiotrophia, which are nutritionally deficient and can cause bacteremia and endocarditis (Senn et al., 2006). Meanwhile, some genera are frequent members of the human microbiota, such as Streptococcus spp. In cases with poor oral health or immunocompromised individuals (example patients with HIV/AIDS, diabetes, cancer, or neutropenia), the genera Campylobacter, Corynebacterium, Eikenella, Haemophilus, Peptostreptococcus, Streptococcus, Treponema, Capnocytophaga, Prevotella, and Fusobacterium can cause opportunistic oral infections (Sedghizadeh, Mahabady & Allen, 2017).

Several investigations have demonstrated the involvement of oral bacteria in inducing oral diseases, including tooth decay (Streptococcus mutans, Streptococcus sobrinus, and Lactobacillus spp.), endodontic infections (F. nucleatum, Enterococcus faecalis, and Propionibacterium spp.), periodontal diseases (Porphyromonas ginigvalis, Tannerella forsytia, Treponema denticola, and Aggregatibacter actinomycetemcomitans), oral cancers (P. ginigivalis and Streptococcus gordonii), and oral lichen planus (Capnocytophaga sputigena, Eikenella corrodens, and Prevotella intermedia) (Nagarajan, Prabhu & Kamalakkannan, 2018; Prada et al., 2019; Zargar et al., 2019).

Mycobiome

The fungi/yeast are other main constituents of the oral microbiome. To date, 154 species of fungi in 81 genera and five phyla (Basidiomycota, Ascomycota, Chytridiomycota, Glomeromycota, and unclassified) have been reported in the human oral mycobiome (Peters et al., 2017), of which Candida species are the most significant and abundant oral fungal species (found in 70% of healthy people) (Ghannoum et al., 2010). Candida albicans is the most frequent species (occurring in 40–80% of healthy people), followed by C. glabrata, C. parapsilosis, C. krusei, C. tropicalis, C. kefyr, C. metapsilosi, C. stellatoidea, and C. khmerensis (Radaic & Kapila, 2021). Besides, Candida species are the causative agents of various oral infections. For example, Candida albicans can form biofilm on solid surfaces. This ability helps this fungus invade the adjoining cells and infect them (Sharma et al., 2018). Moreover, C. dubliniensis has been recognized in the oral ulcers of periodontal patients (Nagarajan, Prabhu & Kamalakkannan, 2018). The additional fungal genera related to the oral cavity are Aureobasidium, Cladosporium, Malassezia, Aspergillus, Saccharomycetales, Cryptococcus, and Fusarium (Radaic & Kapila, 2021; Deo & Deshmukh, 2019). Even though the three species Cryptococcus, Aspergillus, and Fusarium are pathogenic for humans, the interactions between oral fungal commensals and these species may control their pathogenicity (Ghannoum et al., 2010). Among the reasons for the scarcity of studies on mycobiome are the relative rarity of fungi compared to oral bacteria (<0.1% of the oral microbiome, according to the CFU), the inability to culture several fungal species with existing culture techniques, the difficulty of extracting the fungal genome, the lack of databases specific to fungi, and, as a result, the struggle of analyzing their genome sequences (Baker et al., 2017). With advances in technology, including shotgun metagenomics, more research should be conducted to clarify the function of fungi in the oral cavity and to distinguish the true oral mycobiome from transient species (Radaic & Kapila, 2021).

Archaeome

Archaea are other components of the oral microbiota in humans (Verma, Garg & Dubey, 2018), including the members of five methanogenic genera: Methanobrevibacter, Methanosphaera, Methanosarcina, Thermoplasmata, and Methanobacterium. Among them, three species, Methanobrevibacter oralis (40%), Methanobacterium curvum/congolense, and Methanosarcina mazeii, are the most prevalent oral archaeome. Archaea are less frequent and disparate than bacteria, and their distinct role in oral health has yet to be identified. Several investigations verified the higher prevalence of archaea (especially M. oralis) in periodontitis, root canal necrosis, and peri-implantitis. The ability to form biofilm and interact with immune cells could potentiate archaea in establishing oral infections. Other research revealed the coexistence of archaea with some oral pathogens, such as Treponema denticola, P. gingivalis, and Tannarella forsythia. Archaea are probably the final degraders of their host’s components and thus cause the continuation of the catabolic cascade, which is effective in causing oral diseases (Radaic & Kapila, 2021).

Protozoome

Previous research has confirmed that protozoa species, including Entamoeba gingivalis and Trichomonas tenax, are mouth parasites. However, later studies refuted this claim and showed their presence in healthy people as oral microorganisms. These two species are the most abundant oral protozoa (Radaic & Kapila, 2021; Deo & Deshmukh, 2019). Gingival tissues adjacent to the teeth, gums, and rarely tonsils are the natural habitats of E. gingivalis. These nonpathogenic protozoa feed on bacteria, food debris, and mouth epithelial cells (Nagarajan, Prabhu & Kamalakkannan, 2018). Their prevalence is higher in people with poor oral hygiene and those with periodontal diseases and gingivitis. This higher prevalence is due to the increased number of nutrients the bacteria provide (such as P. gingivalis, T. denticola, and Eubacterium nodatum) present in the infection site. Studies showed that most of the oral protozoome are saprophytes. The exact function of protozoa in the mouth has yet to be precisely identified and requires further study (Martínez, Kuraji & Kapila, 2021).

Virobiome

The human virome or virobiome consists of different kinds of viruses, from eukaryotic viruses that infect human cells, to bacteriophages (prokaryotic viruses) that are bacteria- or archaea-specific viruses (Liang & Bushman, 2021), and human endogenous retroviruses (HERVs) (Rascovan, Duraisamy & Desnues, 2016), which are incorporated into the human genome and comprise about 8% of it. A summary of different components of the human virobiome is shown in Fig. 4 (Bai et al., 2022). The total number of viruses in the human body is approximately the same as bacterial and human cells (Shkoporov & Hill, 2019). Studies revealed that healthy individuals carry a diverse population of eukaryotic and prokaryotic viruses (Baker et al., 2017), and bacteriophages have a higher abundance (Liang & Bushman, 2021)

Figure 4 Summary of different components of the human virobiome.

Adapted from Roux, Matthijnssens & Dutilh (2021).

Human oral virome has a very conserved and individualized composition, which is sex-dependent (Liang & Bushman, 2021; Abeles et al., 2014). It showed that there were sex-based differences observed in bacteria to communities of human viruses. There was a significantly greater proportion of shared homologs within each sex (31.1) for males and 34.3 for females when compared with the proportion between sexes (Abeles et al., 2014). Some of the most frequent eukaryotic viruses found in healthy oral virome are Anelloviridae (the most prevalent), Papillomaviridae, such as Human Papillomavirus (HPV), Herpesviridae, such as Human Cytomegalovirus (CMV), Herpes simplex virus type-1 (HSV-1), Epstein-Barr virus (EBV), and Redondoviridae (Baker et al., 2017). Primary research suggests a link between increased levels of redondoviruses and some medical issues, such as respiratory diseases and periodontitis (Abbas et al., 2019; Spezia et al., 2020). In addition, members of the herpes virus family, such as the HSV, EBV, and CMV, are competent at rendering oral diseases. For example, HSV-1 and HSV-2 provoke cold sores, a recurrent oral ailment. EBV and CMV cause a crucial disease called mononucleosis that is transmitted via oral contact. In addition, human papillomaviruses have a role in inducing clinical manifestations such as papilloma, condyloma, and focal epithelial hyperplasia, thus affecting oral health (Nagarajan, Prabhu & Kamalakkannan, 2018). Nevertheless, the function of most eukaryotic viruses in the oral microbiota is still undetermined and requires further research.

Bacteriophages are the principal members of the oral virobiome (Rascovan, Duraisamy & Desnues, 2016). They have constant lysogenic and lytic cycles that enable them to affect the oral bacteria and cause changes in bacterial communities (Nagarajan, Prabhu & Kamalakkannan, 2018).

Bacteriophages

Bacteriophages, also known as phages, are viruses that mainly infect bacteria. They also have the ability to infect archaea, which are other prokaryotic microorganisms (Ly et al., 2014). Commonly, viruses of the family Lipothrixiviridae are specific to archaea (Rascovan, Duraisamy & Desnues, 2016). The vast majority of archaeal phages infect extremophilic hosts, have a broad diversity regarding virion shapes and are distinct from eukaryotic and bacterial viruses. Bacteriophages are diverse and broadly distributed in the human body (Liang & Bushman, 2021; Azam et al., 2021; Pelyuntha et al., 2022). They have diverse species and the lifecycle of bacteriophages is another factor that distinguishes them from eukaryotic viruses. These bacterial predators have two different lytic and lysogenic lifecycles, which regulate the function and biodiversity of bacterial populations and thereby affect the human body’s microbiome and homeostasis (Martínez, Kuraji & Kapila, 2021).

In the lytic cycle, after the bacteriophage genome is injected into the host cell, the phage adopts cell machinery to produce phage particles and lyse the bacterial cell to release the phage progeny. In the lysogenic cycle, the phage genome integrates as a prophage into the host cell genome, delaying virion production. The prophage propagates with bacterial genome replication and distributes among the daughter cells (Szafrański, Slots & Stiesch, 2021; Steier, De Oliveira & De Figueiredo, 2019). Under certain circumstances, such as ultraviolet radiation, DNA-targeting antibiotics, inappropriate pH or temperature, the presence of foreign DNA, and ROS (reactive oxygen species), lysogenic phages change their lifestyle to lytic and lyse the bacterial cell to spread new virions (Szafrański, Slots & Stiesch, 2021). Bacteriophages select their lifecycle (lysogenic or lytic) based on the amount of arbitrium (a peptide involved in inter-phage communication) or the presence of the bacterial population, which triggers quorum sensing (Szafrański, Slots & Stiesch, 2021). The amount of lytic and lysogenic bacteriophages can have an impact on the ecological, biochemical, and pathological attributes of the body (Barr, 2017).

Lysogenic or temperate bacteriophages can selectively confer several advantages and new functions to their bacterial host (Abeles et al., 2015), called the lysogenic conversion (Brüssow, Canchaya & Hardt, 2004). For example, phage-encoded proteins can affect bacterial virulence by producing virulence factors (such as toxins produced by Vibrio cholerae, Escherichia coli, and Corynebacterium diphtheriae) or aiding bacterial fitness in the environment (Ly et al., 2014; Zhang et al., 2019). Temperate phages also transfer genes encoding antibiotic resistance, antibody-degrading enzymes, and platelet-binding proteins (pblA and pblB) to their bacterial host (Girija & Ganesh, 2022), In addition, they may act as reservoirs of virulence genes involved in extra-oral colonization and immune evasion (Szafrański, Winkel & Stiesch, 2017). Moreover, these phages can protect bacterial cells against lytic bacteriophages through the induction of superinfection-related immunity (Barr, 2017). Temperate phages also involve bacterial horizontal gene transfer (HGT) through transduction (Zhang et al., 2019), which is the transferring of genetic materials between bacteria by using phages as a vehicle (Edlund et al., 2015).

It was mentioned that the healthy human oral phageome generally comprises three families of the order Caudovirales (tailed phages), including Siphoviridae (lysogenic phages with long and non-contractile tails and intermediate host ranges), Podoviridae (lytic phages with short and non-contractile tails and moderately extensive host ranges), and Myoviridae (lytic phages with contractile tails and relatively narrow host ranges) (Liang & Bushman, 2021; Ly et al., 2014; Zhang et al., 2019). The morphological characteristics of the three main bacteriophage families in oral virobiome is shown in Fig. 5 (Martínez, Kuraji & Kapila, 2021). In a study published in 2023 by Turner et al. (2023), significant changes to the taxonomy of bacterial viruses: the paraphyletic morphological families Podoviridae, Siphoviridae, and Myoviridae plus the order Caudovirales were abolished. In addition, numerous integrase genes (involved in the lysogenic cycle) have been detected among oral phages. These results suggest a higher abundance of temperate phages in the oral cavity and a high rate of lysogenic conversion of oral bacteria (Edlund et al., 2015), which leads to a dynamic balance with related bacterial hosts (Baker et al., 2017). Alongside the profound function of temperate bacteriophages, the lytic phages play an essential role in adjusting the oral bacteriome. They cause 20–80% of the bacterial death, consequently limiting bacterial growth (Radaic & Kapila, 2021).

Figure 5 (A–C) The morphological characteristics of the three main bacteriophage families in oral virobiome.

Adapted from Martínez, Kuraji & Kapila (2021).

Oral phages are generally safe for human cells (Szafrański, Slots & Stiesch, 2021), and one of their main characteristics is persistence in the mouth. Abeles and colleagues (Abeles et al., 2015) evaluated the salivary phageome of eight individuals during a 60-day time interval and revealed that almost 20% of the oral phages were stable during this time. In addition, they analyzed a single phage throughout the study and observed only a few polymorphisms at the gene level, which shows the genomic persistency of oral phages (Abeles et al., 2014). These microorganisms use numerous methods to evade the immune system and persist in the oral cavity, such as having their own specific restriction or modification enzymes and preventing similar sequences for restriction or modification systems. These enzymes help oral phages modify their nucleic acids, mimic bacterial hosts, or broaden the phage host range (Tock & Dryden, 2005).

Metagenomic analysis for the study of oral bacteriophages

Studies by researchers over centuries showed the role of specific viruses in causing various diseases; however, conventional methods such as culturing, serological identification, and microscopic examination could not fully determine the diversity and function of the viral populations (Szafrański, Slots & Stiesch, 2021). With the advancement of DNA sequencing technologies and the improvement of analytical capabilities, the detection of viruses was facilitated, and as a result, the knowledge of viral communities dramatically increased. A non-targeted sequencing method called “shotgun metagenomics” can be done to investigate pure samples of the environmental virus population. This technique eliminates the need for cultivation and instead relies on directly sequencing viral genomes extracted from a sample. The sample purification process involves a combination of different methods, including filtration, precipitation, and DNase/RNase treatment. The ultimate samples contain encapsulated fractions of viral DNA or RNA, which can be used for sequencing (Roux, Matthijnssens & Dutilh, 2021). Using this method, they could identify large amounts of viral dark matter (formerly non-characterized viruses) and highly abundant and diverse bacteriophage genomes. This approach has been used in numerous studies on virobiomes, revealing the function of the human virobiome in health maintenance and causing diseases and highlighting the importance of viral dark matter (Liang & Bushman, 2021; Bai et al., 2022). Shotgun metagenomics method identifies microorganisms through sequencing of the entire sample nucleic acid contents. In addition, this approach can be employed for strain identification, predicting antibiotic resistance, and evolutionary tracing (Bai et al., 2022). This technique has the potential to determine the functional capability of the microbiome, uncover novel enzymatic functions and genes, comprehend the interactions between host and pathogens, and discover new healing approaches to human illnesses (Bikel et al., 2015).

Viral metagenomics (also called viromics) is a valuable means to describe various viruses, including bacteriophages (Zhang et al., 2019). This technique overwhelmed the principal boundaries of the conventional methods employed for virus detection, which needed the exact genetic information of formerly isolated or described viruses (Rascovan, Duraisamy & Desnues, 2016). New phages can be recognized by comparing unknown sequences with known ones in the database (Zhang et al., 2019). Recently, several viral databases have been expanded, such as the Human Virome Database (HuVirDB), Cenote Human Virome Database (CHVD), Gut Virome Database (GVD), Gut Phage Database (GPD), and Metagenomic Gut Virus (MGV) catalog. These databases showed the enormous human viral variousness and have contributed to the progress of viral studies for determining their characteristics and interactions with host cells (Li et al., 2022). Some advantages of phage metagenomics include obtaining an overview of the phage population and lifestyle, identifying species targeted by phages, and discovering their genetic information without phage isolation, such as the presence of toxins. In addition, it provides the possibility of investigating the potential for phage engineering and describing phage populations in intricate clinical states. In addition, screening bacterial genomes using metagenomics helps to detect CRISPR spacers and phage-like elements (Szafrański, Slots & Stiesch, 2021; Szafrański, Winkel & Stiesch, 2017). Moreover, profiling oral biofilms with metagenomics techniques increased our understanding of the possible function of bacteriophages in the progress, control, and management of oral infectious diseases (Szafrański, Slots & Stiesch, 2021).

There are several challenges to studying bacteriophages through metagenomics. The viral metagenomics differs from bacteria as bacteriophage genomes do not have 16S rRNA sequences and the lack of a marker gene in phage genomes confounds their identification and taxonomy studies (Bai et al., 2022; Szafrański, Slots & Stiesch, 2021). In addition, the effect of sequencing errors on the final detection of viruses is significant and can lead to false identification (Zaura et al., 2014). Despite the dramatic development of virus-specific databases, the number of deposited phage genomes is still low, and most new sequences need to be better annotated. Another limitation of the metagenomics technique is the presence of host DNA in the samples, which interferes with identifying phage sequences. Even if the host DNA is absent in the sample, metagenomics requires many target sequences (Bikel et al., 2015).

The extracted viral genome has a low quantity, so it should be amplified to obtain sufficient viral DNA. Many methods have been developed for this purpose, including Multiple Displacement Amplification (MDA), random amplified shotgun library (RASL), and linker-amplified shotgun library (LASL) (Bikel et al., 2015). Present protocols for producing viral metagenomes use the concentration or purification of circulating viral particles by centrifugation and filtration to remove contaminating unencapsulated nucleic acids via nucleases. Therefore, these protocols are highly sample-dependent. This limitation leads to the loss of latent viruses (proviruses and prophages) that exist in host cells for a long time (as incorporated in the host genome as prophages or extra-chromosomal forms) and possibly results in an undervaluation of the function or diversity of these viruses (Rascovan, Duraisamy & Desnues, 2016). Despite all the challenges of this method, the use of metagenomics techniques to study bacteriophages is expanding, and researchers utilize this method to determine the variety and structure of phage populations in the environment and the human body.

Bacteriophages in the healthy oral cavity

The significance of human oral virobiome, particularly phageome, has enticed the attention of researchers in the last few decades. In a recent metagenomics study (Yahara et al., 2021), saliva samples of healthy volunteers were assessed by Illumina HiSeq (short-read sequencing) and PromethION (long-read sequencing). These two sequencing methods detected hundreds of new viral contigs (Table 1). They also recognized nine jumbo bacteriophages (phages with genomes >200 kb) and one Streptococcus bacteriophage group. Moreover, homologues of antibiotic-resistance genes such as beta-lactamases are present in a high proportion of phages (67% of the jumbo phages and 86% of the bacteriophage group). High diversity was identified among oral bacteriophages and uncovered their ability to evade CRISPR-mediated immunity. In addition, it showed that the PromethION sequencing method is efficacious in discovering oral phages and their functions.

In another study conducted in Spain (Pérez-Brocal & Moya, 2018), oral wash samples from 72 healthy students were collected and analyzed using a metagenomics approach. After bioinformatics analysis, 1,339,784 bacterial reads and 204,057 viral reads were identified. Among the viral sequences, 92% were associated with prophages, and only 8% were described as lytic bacteriophages and eukaryotic viruses. The identified prophages belonged to the bacterial genomes of the families Streptococcaceae, Neisseriaceae, Fusobacteriaceae, and Veillonellaceae (n = 71). Most of the bacteriophages belonged to the Siphoviridae (n = 71), Myoviridae (n = 68), and Streptococcus phages (n = 69) (Carr et al., 2020). HHV-7 (n = 61) and the other members of the Herpesviridae were the most prevalent eukaryotic viruses. They found that the oral viruses are similar regardless of geographic location. Similarly, Willner et al. (2011) determined the oropharyngeal viral communities of healthy people with metagenomics. Analysis of samples demonstrated the presence of large quantities of bacteriophages and a small amount of Epstein-Barr virus in the oral virobiome. Several phages were identified, such as Propionibacterium acnes phage PA6, Escherichia coli phage T3, and Streptococcus mitis phage SM1. The viral populations had a low distribution, and the estimated abundance of virobiome was 236 species. In addition, this was the first report of the presence of pblA and pblB genes in phage SM1 in the mouth. The pblA and pblB proteins play a substantial role in the binding of S. mitis to platelets and are essential virulence factors of this bacterium. Therefore, phage SM1 has an influential role in S. mitis virulence. The mouth is a rich source of phage SM1 and genes encoding platelet-binding proteins and suggests the presence of a variable, complicated, and inter-individual viral profile that is affected by environmental factors (Carr et al., 2020; Willner et al., 2011; Pride et al., 2012).

Table 1 The number and ratio of viral sequences recognized in each specimen and stratified by the “likely” and “most confident” prophages and phages (adapted from Yahara et al., 2021).

Sample	Phage	Prophage	
	Most confident	Likely	Most confident	Likely	
	Novel	Known	Novel	Known	Novel	Known	Novel	Known	
1	0 (0%)	5 (100%)	54 (59%)	37 (41%)	26 (46%)	30 (54%)	233 (74%)	83 (26%)	
2	0 (0%)	7 (100%)	37 (49%)	38 (51%)	27 (56%)	21 (44%)	205 (72%)	81 (28%)	
3	7 (44%)	9 (56%)	63 (52%)	58 (48%)	19 (37%)	33 (63%)	323 (77%)	97 (23%)	
4	1 (20%)	4 (80%)	25 (42%)	35 (58%)	3 (12%)	21 (88%)	73 (56%)	58 (44%)	
Notes.

The novel phages and prophages do not cluster with any viral sequence in the IMG/VR v2.0 database.

Bacteriophages in oral diseases

Periodontitis, or periodontal disease, is the inflammation of tooth-surrounding structures such as the periodontal ligament, gums, and alveolar bone. It is one of the most frequent dental infectious diseases (Nagarajan, Prabhu & Kamalakkannan, 2018; Ly et al., 2014) and typically occurs when the oral microbial population increases and dramatic changes happen in the composition of the oral bacterial community (Prada et al., 2019). Researchers believe that the existence of particular pathogens in the mouth triggers the host’s immune responses, resulting in periodontitis occurs (Szafrański, Winkel & Stiesch, 2017). So far, the role of some bacterial and viral species in developing periodontitis has been established, including F. nucleatum, P. gingivalis, T. forsythia, S. mutans, A. actinomycetemcomitans, E. nodatum, and T. denticola (Roberts & Mullany, 2010; Rascovan, Duraisamy & Desnues, 2016) as well as EBV, HSV-1, and CMV. However, the effect of other microorganisms, such as bacteriophages, is obscure (Szafrański, Winkel & Stiesch, 2017). Dental caries is another infectious tooth disease caused by the disruption of tooth structure by acids created by oral bacteria during carbohydrate fermentation. Acid disrupts the equilibrium between tooth minerals and oral biofilms. It facilitates the growth of aciduric bacterial species, including Lactobacillus and Streptococci genera members, leading to increased acid production and tooth demineralization (Rascovan, Duraisamy & Desnues, 2016). Regarding the effects of bacteriophages on the structure of bacteriomes, the content and assortment of oral phageome, and their influence on bacterial pathogens, and the progress of oral diseases were questionable. In recent years, advanced techniques such as metagenomics have come to the aid of researchers to investigate the diversity and role of bacteriophages in the development of oral illnesses.

A metagenomics study was performed in China to discover the structure and diversity of the oral phageome and possible interactions between phages and bacteria. They enrolled 40 human subjects (10 periodontitis patients and 30 periodontally healthy controls) (Wang, Gao & Zhao, 2016). The samples examined in this study were 20 dental plaques, 10 of which were from periodontitis patients and 10 from healthy subjects, in addition to 20 saliva samples from healthy individuals. One hundred four unique bacteriophages belonging to the order Caudovirales and the families Myoviridae, Siphoviridae, and Podoviridae were identified. The predicted phages were categorized as the species Lactococcus, Mycobacterium, Actinomyces, Streptococcus, Corynebacterium, Pseudomonas, and Yersinia bacteriophage. Meanwhile, Streptococcus and Pseudomonas phages showed the most diversity. The comparison of the Shannon–Wiener diversities and Bray-Curtis dissimilarities indices indicated a large variety in the composition of phages and bacteria associated with healthy people. However, in periodontitis patients, dental plaques contained a homogenous and similar population of bacteria and phages. These bacteriophages, capable of cross-infection, had a positive association with commensals but a negative relationship with the chief pathogens. This fact suggests a potential association between these phages and the structure of the oral microbiome.

In 2018, researchers from the USA performed a metagenomics-based study with a high-resolution analysis to explore supra-gingival microbiota in 30 children (Al-Hebshi et al., 2019). A whole-mouth, supra-gingival plaque specimen was taken from each child and analyzed by ion torrent sequencing technology. Amongst the resultant sequence reads, bacterial sequences comprised 99.6% of total reads related to 726 bacterial strains classified into 12 phyla, 94 genera, and 406 species. Moreover, two protozoa, 34 phages, and two fungi were detected. The core bacteriome phyla consisted of Actinobacteria (46.7%), Firmicutes (22.5%), Bacteriodetes (14.5%), Proteobacteria (5.8%), Fusobacteria (5.8%), Saccharibacteria (4%), and Spirochetes (0.54%). The most abundant bacterial genera were as follows: Actinomyces (36.05%), Streptococcus (8.4%), and Capnocytophaga (6.1%). Overall, 217 to 301 bacterial species/strains per sample were identified. Some of the identified top core species were Actinomyces spp., Actinobaculum sp., Pseudopropionibacterium propionicum, Corynebacterium matruchotii, and Veillonella parvula. Analysis indicated that the oral microbiota in children with tooth decay differs from that of healthy children regarding the bacterial species. For example, the species of Provetella, Vilonella, Actinomyces, and Atopobium were associated with caries. However, the species of Streptococcus and Leptotrichia were overabundant in the samples of caries-free children. In addition to bacteria, there were differences regarding phageome composition between healthy and caries subjects. The Haemophilus phage HP1 was abundant in children with no caries, and Streptococcus phage M102 correlated to tooth decay. The microbiome of these groups differed functionally. Three deiminases and lactate dehydrogenase were related to a healthy microbiome. However, the microbiome of caries patients was able to produce urate, vitamin K2, and polyamine biosynthesis.

Another cohort study was performed in the USA in 2014, in which dental plaques from supra-gingival and sub-gingival biofilms of 16 participants’ teeth (seven with periodontal disease, nine healthy individuals) as well as their saliva sample were taken and subjected to metagenomics analysis (Ly et al., 2014). Results demonstrated a large community of bacteriophages in all types of specimens. Although various individuals had common viruses in different mouth habitats, for most individuals, virobiome structures were meaningfully related to the oral locations from which samples were obtained. The virobiome composition of sub-gingival and supra-gingival biofilms was significantly associated with oral health; however, this was not true of saliva. The high prevalence of lytic phages of myoviruses observed in sub-gingival plaques signifies the relationship between these phages and periodontal diseases. The high abundance of viruses in periodontal biofilms, particularly bacteriophages, suggests their role in altering bacterial communities in the oral ecosystem and causing periodontitis. Therefore, viruses can show the situation of oral healthiness.

Metagenomics-based studies that compared oral phageome in patients (periodontitis and dental caries) and healthy individuals confirmed noteworthy discrepancies in the diversity and composition of phages in these groups. The phageome of healthy oral cavities was very diverse, but in patients, the oral phageome was homogenous and similar. Most bacteriophages in patients were associated with bacterial pathogens, not commensals. In addition, the functions of the oral microbiome were dissimilar in these groups, which may be associated with phage-related genes and lysogenic conversion. Most studies declared the potential role of bacteriophages in changing the oral bacteriome that resulted in oral diseases through dysbiosis and suggested using these microorganisms as markers of oral health status (Ly et al., 2014; Zhang et al., 2019; Wang, Gao & Zhao, 2016; Al-Hebshi et al., 2019).

Transmission of oral bacteriophages to other individuals

The ecology of the oral microbiome is influenced by sharing our oral microflora with our close contacts. Bacteriophages are no exception to this rule and are shared between close individuals through shared environmental reservoirs or personal contact (Rusin, Maxwell & Gerba, 2002).

Ly et al. (2016) conducted a cohort study that analyzed the saliva and feces samples of 20 genetically distinct subjects living in different households (eight separate houses containing two individuals and four divergent controls living alone). In each home, a person was treated with antibiotics (amoxicillin or azithromycin) for a week, and the second person obtained vitamin C (placebo) instead of antibiotics. Individuals in the control group did not receive any treatment (antibiotics or placebo). Bioinformatics analysis revealed that considerable amounts of the oral and gut virobiome were identical between genetically disparate, cohabitating persons. Bacteriophages comprised the highest proportion of the virobiome. Everyone had a distinctive pattern of mouth and intestinal bacteriophages compared to their housemates, although they shared a part of their virobiome. In addition, the examination of the different groups that received antibiotic and placebo treatment showed that the fecal viral populations were highly persistent; more than 70% were observable for at least four days, and 30% were detectable between 5 and 6 months later. Antibiotics did not affect the stability of bacteriophages, and identical phage patterns were seen in the groups treated with amoxicillin and azithromycin and the control group. The persistence degree in the intestine was much higher than in the mouth. The results showed the distribution of bacteriophages among households over time, which suggests the transfer of a large part of the microbiome in household contacts.

Finally, the studies clarify the role of bacteriophages in oral health and infections. Apart from the effect on the composition and variousness of the mouth microbiome, the transmission of mouth phages between close contacts results in the dissemination of function-related genes, including complement or immunoglobulin degrading enzymes, as well as antibiotic resistance genes, such as beta-lactamases (Abeles et al., 2014; Ly et al., 2016). Therefore, they will attract researchers in various fields, including microbiologists, dentists, and pharmacologists. They can be employed to improve oral care and treatment protocols or develop new therapeutic approaches, such as bacteriophage therapy. There is a potential role of bacteriophages in changing the oral bacteriome that resulted in oral diseases through dysbiosis suggested using these microorganisms as markers of oral health status.

One of the primary challenges in metagenomics studies is the limited availability of NGS facilities in many countries, along with the high cost associated with analyzing each sample. It is hoped that in the future, as sequencing facilities become more widely available and costs decrease, metagenomics studies with larger sample volumes will be conducted more extensively in various parts of the world. Furthermore, developing sequencing technologies can significantly enhance the speed and accuracy of sequencing results, ultimately improving metagenomics outcomes. In addition, as the number of phage genomes registered in databases increases, the identification of phages is made easier by metagenomic methods.

Conclusions

Metagenomics analyses help to present the variety and function of the human microbiome, particularly oral phages, and define their role in sustaining health or developing diseases. There is a substantial disparity in the population of oral phages between healthy individuals and those with oral infections. Healthy individuals have a higher proportion of prophages, which serve as the source of functional and antibiotic resistance genes and are more related to commensal bacteria; however, patients’ phages are associated with pathogenic bacteria, which can cause dysbiosis and change the population of oral bacteria.

Given the results highlighting the significance of bacteriophages in the oral microbiome of healthy and diseased individuals, it is anticipated that regulatory systems will implement strategies to leverage this information for enhancing preventive and therapeutic approaches to oral and dental diseases. Furthermore, it is hoped that in the future, extensive metagenomic studies will be conducted on the oral phageome of diverse individuals, including immunocompromised patients. Conducting these studies play a supportive role in identifying the phageome and, consequently, the oral bacteriome of these patients, ultimately contributing to the enhancement of their health and treatment.

Additional Information and Declarations

Competing Interests

Author Contributions

Data Availability

Dinesh Rokaya is an Academic Editor for PeerJ.

Maryam Banar conceived and designed the experiments, performed the experiments, analyzed the data, prepared figures and/or tables, and approved the final draft.

Dinesh Rokaya analyzed the data, prepared figures and/or tables, authored or reviewed drafts of the article, and approved the final draft.

Reza Azizian conceived and designed the experiments, performed the experiments, analyzed the data, prepared figures and/or tables, and approved the final draft.

Zohaib Khurshid analyzed the data, prepared figures and/or tables, authored or reviewed drafts of the article, and approved the final draft.

Morteza Banakar conceived and designed the experiments, performed the experiments, analyzed the data, prepared figures and/or tables, authored or reviewed drafts of the article, and approved the final draft.

The following information was supplied regarding data availability:

This is a literature review.

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
