# Peer review of "Oral bacteriophages: metagenomic clues to interpret microbiomes"

_PeerJ, doi:10.7717/peerj.16947_

## Round 0.1 · original submission · Major Revisions

Please take good care in addressing both reviewers commentaries, particularly reviewer 2 concerning the references for your table, as well as the discussion.

**Language Note:** The review process has identified that the English language must be improved. PeerJ can provide language editing services - please contact us at [email protected] for pricing (be sure to provide your manuscript number and title). Alternatively, you should make your own arrangements to improve the language quality and provide details in your response letter. – PeerJ Staff

·

Basic reporting

The authors make a valuable contribution to the field, particularly by shedding light on the role of phages in the oral microbiota, a facet that has not been extensively addressed in previous reviews of the oral microbiome. The review addresses a topic of broad, interdisciplinary interest, which fits well with the scope of the journal. The introduction effectively introduces the topic, describes the intended audience, and explains the motivation for the research. The literature review provides ample information on the topic. However, I would like to suggest some improvements in the English language to improve clarity for an international audience. Specifically, I have identified instances in lines 31, 33-34, 38-39, 64-66, 294, 313-315, 395-396, 511-512 where the current wording may hinder comprehension. I recommend careful rewriting of these sections to ensure that the content is more accessible to a global audience.

Experimental design

The authors provide a comprehensive and unbiased coverage of the subject matter, and the organization of the review into logically coherent paragraphs contributes to the overall clarity and flow of the manuscript. However, I would like to suggest a careful review of the author instructions to ensure that in-text citations and references follow the correct format specified in the guidelines.

Validity of the findings

The authors present a well-developed and well-supported argument that effectively aligns with the objectives outlined in the Introduction. Although the manuscript adequately addresses the limitations and outlines future perspectives throughout the review, I would like to suggest that it would be beneficial to include a section devoted to these aspects in the conclusions.

Additional comments

I recommend a careful review of the style considerations outlined in the Author Instructions to ensure consistency in the formatting of taxonomic classifications, including species, genus, family, order, class, phylum, and so on. For example, in lines 444, 446, 489, and 502-507.

Line 59, 64, 70, 144, 152, 256. Microflora may not be the most appropriate term, as it alludes to the kingdom Plantae.

Line 198-204. It would be useful to have a brief generic description of the type of organisms covered by the genera listed.

Line 239-240. Check the use of italics. It is possible that the genera may need to be italicized.

Line 276-277. It would be interesting to mention the way in which the human oral virome is sex-dependent.

Line 281, 286, 548. Check the use of italics. Italics may not be required here.

Line 283. Herpes simplex virus name may need to be capitalized.

Line 295. As written, it appears that there are some bacteriophages of the family Lipothrixiviridae that sometimes infect archaea. Viruses belonging to the family Lipothrixiviridae are actually archaea-specific viruses. I recommend reading Krupovic et al., 2018 (https://doi.org/10.1016/j.virusres.2017.11.025) to clarify the idea and rewriting the sentence.

Line 298. It would be helpful if the authors clarify if they mean 1,015 unique species of bacteriophages in the human body.

Line 298-299. Check the use of italics. Italics may not be required here.

Line 311. I suggest using a different choice of words to make the sentence sound less anthropomorphic.

Line 323. Check the use of italics. Italics may not be required here.

Line 330. Order Caudovirales, as well as families Siphoviridae, Myoviridae and Podoviridae have been abolished. See Turner et al., 2023 (https://doi.org/10.1007/s00705-022-05694-2). I suggest adding this clarification in the text.

Line 343. It would be helpful to have a reference here to support these assertions.

Line 346. Please make sure to correctly cite Abeles and colleagues.

Line 362-364. The authors say that shotgun metagenomics can be used to study “pure samples” of the environmental virus population. It is not clear if they mean a pure culture, purified DNA from a complex sample or viral DNA from a filtered sample. Roux, 2021 (https://doi.org/10.1016/B978-0-12-809633-8.20957-6) may help clarify the concepts of shotgun metagenomics and viromics.

Line 403-404. It would be helpful to have a reference here to support these assertions. It may be appropriate to rephrase the sentence to clarify whether the authors are referring to DNA concentration.

Line 429, 445, 454, 455, 534-535. Please make sure to write the virus names in an appropriate style. For instance, “Escherichia coli” may be required to be italicized in “Escherichia coli phage T3”.

Line 467-468. As written, prophages are understood to belong to the bacterial families mentioned (as if they were bacteria). I suggest modifying the sentence to avoid any confusion.

Table 1. There is no reference 70.

Reviewer 2 ·

Basic reporting

This review article "Oral bacteriophages: metagenomic clues to interpret microbiomes" is indeed of broad and cross-disciplinary interest, aligning with the scope of PeerJ. It provides a comprehensive background and sufficient literature references, which indicates it's not a redundant contribution to the field. The introduction effectively introduces the subject, clearly defining the audience and the motivation for the review. This suggests it adds value to the existing body of research by offering a unique perspective or making the topic accessible to a different audience.

Experimental design

The review survey section is well designed.

However, the survey methodology section should list all the 66 articles in a table in supporting information with statistics about the features of these 66 articles. Moreover, it is important to mention the detailed criteria of why the authors chose these 66 articles.

While the paper reviews articles from a span of over two decades (2000 to 2022), it might benefit from including a discussion on how advancements in technology and methodologies over this period could affect the comparability and relevance of the studies reviewed. For example, in section Metagenomic analysis for the study of oral bacteriophages (line 356), please mention how the development of metagenomics sequencing technique, bioinformatics analysis tools can further help this field.

The section "Bacteriophages in the healthy human oral cavity" in line 419 has only 3 papers, each of which was covered by one entire paragraph. May you shorten it and focus more on the overall summary and discussions (1-2paragraphs) to compare these different studies?

Beyond the reviews about the relevant topics, please include one paragraphs to summarize and discuss future directions of this field. For example, what types of technical challenges can be tackled, and if that is resolved, what novel discoveries can be found?

Validity of the findings

Overall, the arguments are well developed to meet the goals in the introduction. But the conclusion section failed to clearly convey the information about the unresolved questions, gaps, future directions.

---

## Round 0.2 · accepted · Accept

I have carefully reviewed the authors' corrections and I believe they addressed all the reviewers' concerns.